


# Subsoil seismic characterization through Vs30 for future structural assessment of buildings (Ciudad del Carmen, Mexico)

Leonardo Palemón-Arcos[1], Carmen M. Gómez-Arredondo[2], Daniel A. Damas-López[2], Guillermo Chávez-Hernández[2], Yuriko Gutiérrez-Can[1], Marco A. Hernández-Hernández[1], Edén Bojórquez[3]. Francisco Barrera-Lao[4]

[1]Departamento de Ingeniería Civil, Universidad Autónoma del Carmen, Cd. del Carmen, Campeche 24180, México.
[2]División Académica de Ciencias Básicas Unidad Chontalpa. Universidad Juárez Autónoma de Tabasco. Cunduacán, Tabasco 86690, México.
[3]Facultad de Ingeniería, Universidad Autónoma de Sinaloa, Calzada de las Américas y B. Universitarios s/n, C.P. 80040, Culiacán, Sinaloa, México.
[4]Universidad Autónoma de Campeche. Facultad de Ingeniería y Contaduría. Campeche. México.

*Correspondence to*: Leonardo Palemón-Arcos (lpalemon@pampano.unacar.mx) and Carmen M. Gómez-Arredondo (carmen.gomez@ujat.mx)

**Abstract.** Although the seismic information from the subsoil is very important, in some areas of the world this is not available due to various factors, the main one being a seismically low area. It is important to say that the planet has been changing and many intraplate earthquakes have occurred in places never expected, spreading seismic waves to places where they were considered low seismicity. For example, on September 8, 2017 in Ciudad del Carmen, 500 km from the epicenter, the earthquake was felt causing damage to the facades of the buildings. Therefore, it is important to have the subsoil shear-waves velocities to subsequently generate a good analysis and structural seismic design. For this reason, in this study under the seismic approach an assessment of Ciudad del Carmen Campeche subsoil is presented. Active and passive Multichannel Analysis of Surface Waves and Refraction Microtremor technique to investigate seismically subsoil characteristics have been employed. Shear wave velocities were obtained up to a depth of 30 m with magnitudes of 172.45 m/s to 353.90 m/s. Based on the Vs30 values, the subsoil is seismically classified into D and E according to the criterion of the National Earthquake Hazards Reduction Program and International Building Code, turning out to be very vulnerable to high damage during the earthquake shaking. Furthermore, Ciudad del Carmen was regionalized into three types, where type I being a dense soil or averagely soft rock with Vs30 greater than 360 m/s, type II when the soil has an intermediate dynamic amplifications with Vs30 between 180 to 360 m/s, and type III correspond to a soil with large dynamic amplifications and Vs30 less than 180 m/s.

## 1 Introduction

Earthquakes are one of the most severe natural disasters on earth and wreak terrible destruction, generating many losses of human life and heritages property assets (Zhu *et al*., 2019). Earthquakes are natural phenomena product of the interaction and release of huge amounts of stored energy. Due to the elasticity and deformation of the rocks, the sudden and violent shaking





occurs, travels as seismic waves in all directions through the Earth's layers, reflecting and refracting at each interface. These seismic waves are two main types: body and surface waves. Surface waves consist of Love (Love, 1911) and Rayleigh (Rayleigh, 1885) waves and are almost entirely responsible for structural damage and the destruction buildings but it also

depends on the type of soil in which they are built. The shear-wave velocity average on first 30 m (Vs30) of subsoil, is the suggested standard parameter for estimating surface level response to ground motions (Boore, 2004; Borcherdt, 1994; Dobry *et al*., 2000; Holzer *et al*., 2005; Williams *et al*., 2003). The outline of the paper is as follows. Firstly, to contextualize, an introduction with background and previous studies is included, describing in the first subsection seismology in the world and in Mexico, especially the recent earthquake that occurred on September 2017 that called into question the design codes in the

region and the country (México), while in the second subsection the previous works of seismic characterization of Mexico are written, with the lack of thoroughly regionalizing the region and state. Secondly, the study area which includes location and geological framework are incorporated. Then, the data collection and data analysis are described in the Materials and Methods section. The numerical results considering the wave seismic refraction and wave velocities are presented. Finally, the discussions and conclusions of this paper are given.

### 45 1.1 Background

In the world, there are seven collective properties of Plate Boundaries: (1) Continental Convergent Boundary, (2) Continental Transform Fault, (3) Continental Rift Boundary, (4) Oceanic Spreading Ridge, (5) Oceanic Transform Fault, (6) Oceanic Convergent Boundary and (7) Subduction Zone (Bird, 2003; Morra *et al*., 2013). The Pacific plate is the largest tectonic plate on Earth and its boundaries host most subduction zones and earthquakes (Morra *et al*., 2013). Mexico is surrounded by three

tectonic plate boundaries: (1) a transforming boundary between the Pacific-North American plates, (2) a convergent boundary between the Cocos-North American plates, and (3) a transforming movement between the Caribbean-North American plates (see Fig. 1b) (SSN, 2020). Seismically, Mexico was divided into four regions (CFE, 1993): A, B, C and D, the first being the least seismically and D with the highest spectral parameters. In the 2008 version Chapter of Design by Earthquake of the Civil Works Design Manual of Federal Electricity Commission, the concept of seismic regionalization was eliminated, and site

spectra were considered (CFE, 2008); however, in the new Mexican regulations (CFE, 2015) the old philosophy was returned in which constant acceleration spectra, regional spectra and site specific spectra were considered. The earthquake occurred on September 8, 2017 at 04:49:18 (UTC time) in the Chiapas region (shaded square of Fig. 1b) of southern Mexico (Jimenez, 2018), whose epicenter was located 150 km south of the city of Arriaga at: Lon = −94.11°, Lat = 14.85° and depth of 58 km (SSN, 2020) generated structural damage in Chiapas, Oaxaca, Tabasco and Campeche (Fuentes *et al*., 2019). Therefore, it is

important to know the characteristics of the city subsoil to mitigate the loss of human integrity with the aim of having a seismically resilient city.



### 1.2 Previous studies

For decades, attempts have been made to characterize areas of interest with greater construction by means of spectra for seismic design (Esteva, 1967), as well as site studies with seismic risk for the entire Mexican Republic (Esteva & Ordaz, 1988; Ordaz
*et al*., 1989; Tena-Colunga *et al*., 2009). Currently, the design code shows design spectra by type of soil, which will be constructed from the maximum acceleration in rock, that is, the parameter directly associated with seismic hazard (CFE, 2015; Pérez-Rocha *et al*., 2015). Work has also been done on the optimal values of the plateaus of the design spectra for the limit state of collapse (Rosenblueth, 1976; Esteva & Ordaz, 1988; Rocha & Schroeder, 2008). On the other hand, by using geophysics through seismic and electrical methods, certain states of the Mexican Republic and the world have characterized
their soil through the wave velocities and resistivities of the Earth's surface layers (Alonso, *et al*., 1995; Cosenza *et al*., 2006; Dai *et al*., 2019; Gómez-Arredondo, *et al*., 2016; Gómez-Arredondo, *et al*., 2020; Jaramillo, 2012; Liu *et al*., 2013; Montalvo-Arrieta, 2005; Ozcep *et al*., 2013; Pancha, 2008; Valdebenito *et al*., 2015). All the efforts described in this section have focused on areas of greater infrastructure and with high seismicity, leaving second or third plane the areas of greater infrastructure and low seismicity or of low infrastructure and low seismicity. The present work employs a Multichannel Analysis of Surface
Waves (MASW) and Refraction Microtremor (ReMI) technique to investigate the characteristics of the subsoil and seismically characterize Ciudad del Carmen Campeche Mexico.

### 2 Study area

This section will be divided into two parts, one proper of the geographical location and the other, its ubication between the tectonic plates.

**2.1 Location**

Mexico located in the southern part of United States, is surrounded by the Atlantic and Pacific Oceans (Fig. 1a). The study area is in Ciudad del Carmen, Campeche, Mexico located at 18°38′18″N and 91°50′07″W between the Gulf of Mexico and the Yucatan peninsula (shaded circle of Fig. 1b). Campeche is part of the Mexican southeast and is bordered to the south by the Chiapas state, a wide zone of plate deformation in which three plate boundaries converge: (1) North America, (2) Caribbean
and (3) Cocos plates (Fig. 1b).

**2.2 Geological Framework**

Geologically Ciudad del Carmen of the Campeche state is complex, which is because the movements of the North American, Caribbean and Cocos plates (Fig. 1b) converge in this region from the late Oligocene (Morán Zenteno *et al*., 2000), that is, the North American Plate moves to the west with respect to the Caribbean, while the Cocos Plate moves to the Northeast with
respect to the previous ones. The structures resulting from this tectonic activity during the Mesozoic and Cenozoic have different trends, as well as different deformation ages (Padilla and Sánchez, 2007). The opening of the Gulf of Mexico related





to the rifting event, induced the movement of the Yucatán Block to the south during the Early Cretaceous and Middle Jurassic, due to seafloor spreading. The continental shelf of the Gulf of Mexico presents a lithological change between the recent sediments, contributed by the river system of the Grijalva-Usumacinta rivers, and the marine ones, formed by calcareous

bioclasts (Fig. 1c) (Ayala-Castañares and Gutiérrez-Estrada, 1990).

In the southwest of the Gulf of Mexico, large shallow platforms were also developed that extended to the Massif of Chiapas and the western part of the Yucatan Block that continued to contribute clastics, large volumes of carbonates were deposited there, and probably also some salt deposits in the northwestern part of the Massif de Chiapas (Viniegra, 1971). During this period, high organic matter shales were deposited in the basin, with thin intercalations of carbonates that are the generating

rock of most of the huge volumes of hydrocarbons that exist in the Gulf of Mexico, especially in the Mexican southeast (González and Holguin, 1992). For Oligoceno, the deposit of clastics continued throughout the same area (Ambrose *et al.*, 2003).

The coastal geomorphology of Mexico is primarily shaped by tectonics of the North America, Pacific, Rivera, Cocos and Caribbean plates (see Fig. 1b). The Mexican Pacific coast is parallel to the subduction zone (Ramírez-Herrera *et al.*, 2016)

where the Rivera and Cocos plates subdue beneath the North American continental crust (see Fig. 2). The Rivera plate subdues beneath the states of Jalisco and Colima whereas the Cocos plate subdues beneath Michoacán, Guerrero, Oaxaca and Chiapas. South of Chiapas and in Central America, the Cocos plate subdues beneath the Caribbean plate. Several large subduction earthquakes of Mw > 7 have occurred over the last century (Kostoglodov and Ponce, 1994). Hence, seismic movements within the Mexican Pacific are frequent and its coastline is vulnerable to seismic hazards (Ramírez-Herrera *et al.*, 2016). Of the

interactions described in section 1.1, the study area refers to the Mexican Southeast whose zone is characterized by having a complex tectonic interaction and great potential to generate large earthquakes, see Fig. 2 (*e.g.* January 15, 1931, Mw = 7.8; April 29, 1970, Mw = 7.3; 09/30/1999, Mw = 7.4 and 08/09/2017, Mw = 8.2) (SSN, 2020).

According to the National Seismological Service, from 1974 to 2020 in the Ciudad del Carmen of the Campeche state, seventy-four earthquakes with a magnitude of 3.6 to 4.8 have been presented (Mexico, 2020), 1980 to 1986 being the period of greatest

seismic activity (see Fig. 3).

## 3 Materials and methods

Thirty profiles were made using the MASW technique, taking care to have a free area of at least 150 meters for vertical geophones array with a natural frequency of 4.5 Hz, a PASI GEA® multichannel seismograph with 24-channel (Geometrics, Inc. 2006), seismic cable, laptop computer, 71.2 N heavy duty of sledgehammer, 250 x 250 x 10 mm of steel plate and trigger

sensor to data collect were used. The MASW technique developed by Park, Miller and Xia, (1999) has been widely validated and used in many recent studies in Civil Engineering for microzonation and site response studies (Anbazhagan and Sitharam, 2008a,b; Kanli *et al.*, 2006; Tokeshi, Leo and Liyanapahirana, 2013; Xia, Miller and Park, 1999; Xia *et al.*, 2003) to determinate shear-waves velocities and the dynamic properties of shallow soil profiles (Ivanov *et al.*, 2006; Miller *et al.*, 1999;





Park and Elrick, 1998; Park, Miller and Xia, 1999; Park *et al*., 2005; Park *et al*., 2007; Stewart, Liu and Choi, 2003; Xia *et al*.,
2003). To apply the MASW surveys will carry out three stages: data acquisition, dispersion analysis and shear-wave velocity
inversion, see Fig. 4.

### 3.1 Fieldwork database

For Multichannel Analysis of Surface Waves and Refraction Microtremor techniques applications, the laying of twelve
geophones with a 5 meters separation was carried out, having a total length of 60 meters with three sources positions, two were
positioned of 5 meters from the first and last geophone, and the third one in the middle of array location for each side. The
active MASW and passive Refraction Microtremors methods were used to subsoil survey, see Fig. 5. For the first one, the
seismic energy by hitting the plate three times in each position were generated, so, an elastic wave front will be radiates
outward from the shot point spreading in all directions refracted according to Snell's Law (Fkirin, Badawy and El deery, 2016)
when it impinges on a boundary between two materials with a seismic impedance contrast (see Fig. 5a), whereas the passive
method utilizes surface waves generated passively by twenty minutes ambient noise due to human or natural activities (*e.g.*,
traffic, thunder, waves and tidal motion) per point (see Fig. 5b). The geophones function is to convert the physical movement
of the ground to an electrical signal which related to a clip to the geophone cable. These instruments have a spike on the base
of each geophone ensures adequate physical contact between the geophone and the ground surface. Velocity of and depth to,
the refracting surface can be calculated by measuring the travel time of the seismic wave between the seismic source and the
receivers.

Ciudad del Carmen is an island between Terms Lagoon and Gulf of Mexico and is connected to the mainland by two large
bridges. In 2010 the population was 221,094 (INEGI, 2010). Carmen municipality has ancient and narrow streets with an
Urban Director Program in which are the land use and destination planning, policies by areas and conservation, improvement
and growth actions and urban development policies and guidelines for the prevention of phenomena risks natural and man-
made. Fig. 6 shows the density of infrastructure in the municipality, so, it was decided to extract information at thirty points
in that area.

### 3.2 Data analysis

Rayleigh waves are generated using vertical impacts on the ground surface, which develop coupled P-SV wave energy and
MASW technique is the most common to acquire and process Rayleigh waves. In general terms, dispersion implies a selection
based on a previous separation, in this case, the propagation of surface waves velocities exhibit dispersion due the subsoil
velocities depend on the frequency. Therefore, in the dispersion analysis that they suffer when crossing subsoil layer allows to
infer the characteristics of the medium through which they have traveled. In addition, the crust is formed of heterogeneous
layers and is more stratified than the mantle whose vertical heterogeneity is in the form of an exponential function, a linear
function, a quadratic function (Singh *et al*., 2015), so that, the dispersion curves of Rayleigh waves in vertically heterogeneous
media, can be calculated by Knopoff's method and phase velocity given by the roots of Eq. (1) (Abo-Zena, 1979; Crampin,


1981; Lai and Rix, 1998; Schwab and Knopoff, 1972; Xia, Miller and Park, 1999). In this paper, SeisImager/SW® software package was used to obtained data post-processing *i.e.* dispersion curves (Geometrics, Inc. 2006)

$$\boldsymbol{F}\left[f_j, c_j, v_{si}, v_{pi}, \rho_i, h_i\right] = 0 \tag{1}$$

where $f_j$ is the frequency in Hz; $c_j$ is the Rayleigh-wave phase velocity at frequency $f_j$, $j = 1, 2,\ldots, m$; $v_{si}$, $v_{pi}$, $\rho_i$, and $h_i$ are the

S-wave velocity, P-wave velocity, density and thickness vector, respectively define in Fig. 7 and Eq. (2).

$$v_{si} = \begin{bmatrix} v_{s1} \\ v_{s2} \\ . \\ . \\ . \\ v_{si} \\ . \\ . \\ . \\ v_{sn} \end{bmatrix}, \quad v_{pi} = \begin{bmatrix} v_{p1} \\ v_{p2} \\ . \\ . \\ . \\ v_{pi} \\ . \\ . \\ . \\ v_{pn} \end{bmatrix}, \quad \rho_i = \begin{bmatrix} \rho_1 \\ \rho_2 \\ . \\ . \\ . \\ \rho_i \\ . \\ . \\ . \\ \rho_n \end{bmatrix}, \quad h_i = \begin{bmatrix} h_1 \\ h_2 \\ . \\ . \\ . \\ h_i \\ . \\ . \\ . \\ h_n \end{bmatrix} \tag{2}$$

After the detection of motion on the ground surface, the seismic waves are carried over into the multichannel record and generates accurate imaged through a 2D wavefield transformation. In each survey station of the thirty made, an 2D image from each raw waveform shot record was obtained. In the Fig. 8, two only are shown for each method, active and passive approach.

Notice, in a heterogeneous medium, the surface wave does not have a single velocity but a phase velocity that is a function of frequency.

To transform the waveform data from the space-time domain into the phase velocity-frequency domain SeisImager/SW® was used applying the phase shift method proposed by Park, Miller and Xia (1998, 1999). The dispersion curves *i.e.* phase velocity of Rayleigh waves versus frequency are shown in Fig. 9. In these dispersion curves contribute in a unique way the P-wave

velocity, S-wave velocity, density and layer thickness depicted in Fig. 7; nevertheless, the S-wave velocity is the dominant parameter influencing changes in Rayleigh-wave phase velocity. The dispersion curves were obtained with the accurately possible considering the fundamental mode and using WaveEq module from the same software package.

## 4 Numerical Results

The study of wave propagation in subsoil helps to understand and predict the structure seismic behavior, so, the results of the

thirty points for the active and passive MASW method are shown. For seismic zonation, averages of the velocities of both methods were obtained.



### 4.1 Shear-wave velocities Vs30

Generally, the forward model is presented when the result is obtained from a cause, while the inverse model attempts to seek the cause from a result. Subsequently the inversion, a velocity profile with depth is obtained. The shear-wave velocity is

averaged and is computed using the formula (3) (Borcherdt and Glassmoyer, 1992; Borcherdt, 1994).

$$Vs30 = \frac{\sum_{i=1}^{N} h_i}{\sum_{i=1}^{N} \frac{h_i}{v_{si}}} = \frac{30}{\sum_{i=1}^{N} \frac{h_i}{v_{si}}} \qquad (3)$$

Where $h_i$ is the thickness of the $i^{th}$ layer (in m), $v_{si}$ is the shear-wave velocity in the $i^{th}$ layer (in m/s) and $N$ corresponds to the number of layers identified in the upper 30m of the ground. Fig. 10 shows results obtained at four surveys points using the active technique. The red squares represent the number of blow count at each depth and are factored by a magnitude of 5, the

black circles the unfactored S-wave velocities, while the blue triangles indicate the vertical P-wave velocities divided by a factor of 5.

Surface wave methods are widely used in Civil Engineering to obtain the shear wave velocity profiles of near-surface material, as these methods are longer-range, non-invasive, and low-cost than crosshole/downhole seismic (Bajaj and Anbazhagan, 2019; Foti *et al*., 2009, Socco, Foti and Boierol, 2010). To structurally design a project, geotechnical soil investigation is required

and the Standard Penetration Test according to ASTM (2019) is generally used to obtain the subsoil characteristics where the structure will be built. The Standard Penetration Test (SPT) generate the blows count ($N_{SPT}$) required to drive the sampler over the depth interval of 150 to 450 mm of a 450 mm drive interval using the driving mass in falling free (ASTM, 2019). In geotechnical practice, the SPT has been used in correlations for density, angle of internal friction, undrained compressive strength, stress-strain modulus and it has also been used to estimate the bearing capacity of foundations.

The growth of MASW technique usage, several researchers have correlated the blow count N with shear wave velocity among the main and pioneers stand out Imai and Yoshimura (1970), Ohba and Tourim (1970), Seed and Idriss (1981), and recently Akin, Kramer and Topal (2011), Anbazhagan, Parihar and Rashmi, (2012), Fabbrocino *et al*. (2015), Kirar, Maheshwari and Muley (2016), Tsiambaos and Sabatakakis (2011). In this paper, the blow count and the density widely validated by the mentioned researchers were obtained. The comparison between the active and passive MASW technique is performed

considering three points. These comparisons are presented in S-waves and P-waves velocities, density and blow count, see Fig. 11. A maximum difference between them of around 15% is observed. The thickest solid line corresponds to the active one, while the thin line corresponds to the passive method.

### 4.2 Seismic zonation

From all over the Mexican Republic, only in Mexico City, there is a symbiosis between seismic engineering researchers to

study earthquakes and their effects in that region, forgetting a little about the southeast region in which there have also been significant damages due to earthquakes described in section 2.2. Therefore, to mitigate damage and contribute to having a


resilient city, seismic zonation is carried out with the technique described extensively in section 3 obtaining the results shown in Fig. 12, which shows a small difference of 13% between active (shaded circles), passive (unshaded circles) method in shear-wave velocities average at 30 meters deep. It is observed that the shear wave velocities on the Carmen Island range from

172.45 m/s to 353.90 m/s.

In Mexico, the only design code that considers the shear-wave velocities is the Civil Works Design Manual of Federal Electricity Commission Design - Earthquake Chapter (CFE, 2015). At the international level there are: (1) National Earthquake Hazards Reduction Program, NEHRP (BSSC, 2009) and (2) International Building Code, IBC (ICC, 2017). For seismic design, the NEHRP recommended Seismic Provisions adopted ASCE/SEI 7 standards and in the section 1613.2.2 of IBC, classifies

the site based on the soil properties in A, B, C, D, E and F in accordance with Chapter 20 of ASCE 7. It is observed that the international codes converge in the American Society of Civil Engineers ASCE standard (ASCE, 2017), therefore, the two international codes that relate the site classification with the shear-wave velocities are shown in Table 1. It is observed in the table, that although the main parameter to classify a site is Vs30, strength parameters such as penetration resistance ($N_{SPT}$) and undrained shear strength ($Su$), of the upper 30m of the ground, can also be used.

According to Table 1, Fig. 12 and Fig. 13, Ciudad del Carmen Campeche has a type D and E soil. Based on the Mexican code, a soil type III is observed. Attention should be paid to soil type E because the soil can present liquefaction failure. For major projects classified as group A and A+ (CFE, 2015) a meticulous study subsoil must be carried out in the final project area.

To be in accordance with the Mexican national code whose intention is to derive the Complementary Technical Standards of Carmen municipality, the soil will be regionalized in three types, Table 2 and Fig. 14, where type I being a dense soil or

averagely soft rock with Vs30 greater than 360 m/s, type II when the soil has an intermediate dynamic amplifications with Vs30 between 180 to 360 m/s, and type III correspond to a soil with large dynamic amplifications and Vs30 less than 180 m/s.

## 5 Discussion

The sudden release of the accumulated stresses in an area, generates P, S and surface waves. Of the latter are the Love (Love, 1911) and Rayleigh waves (Rayleigh, 1885). Due to free surface boundary conditions, the waves that damage buildings are

Rayleigh waves, which produce a retrograde elliptical motion of the ground and are the result of the interaction of primary wave (P) and vertically polarized shear wave (SV) motions. The investigation depth for the active method was between 30 to 36 m, while for the passive method it exceeded 40 m, nevertheless, at 30 m the average velocities are representative for the soil-structure interaction design. The seismic zonation was obtained by averaging between the two methods applied in the campaign, generating results with a difference of less than 15%. MASW was used to map the subsoil obtaining, layer

thicknesses, sediment, subsurface interfaces, density and shear wave velocity of each layer. Only in some points the obtained velocity values were correlated with the Geotechnical of the area, which are not shown in this paper but will be published in another article.



According to the CFE seismic regionalization map the study area is classified as a low seismic hazard zone (CFE, 1993, 2008, 2015); however, the shear-wave velocities obtained in this paper are low, which leads to it being a site of can experience large

dynamic amplification of ground motion with the passage of seismic waves. To use Table 1, it will enter using one of the following three data: (1) Vs30, (2) penetration resistance ($N_{SPT}$) and (3) undrained shear strength (*Su*). Care should be taken with soil type E whose shear-wave velocities are below 182 m/s, because these soils vulnerable to potential failure or collapse under seismic loading, such as liquefiable soils and collapsible weakly cemented soils (BSSC, 2009).

## 6 Conclusion

A fault in tectonic plates generate seismic waves that propagate in all directions through the subsurface layers, and the damage caused in a particular surface site, is strongly dependent on the source mechanism, source depth and the characteristics of rock and soils types (*e.g.*, subsoil velocities or peak ground acceleration) along its travel path. The latter is so-called local site effects and generally not all regions of the world have this seismic information. Therefore, this paper shows the seismic regionalization through Vs30 for future seismic structural analysis and design with soil-structure interaction or with seismic isolation whose

necessary data are the shear-wave velocities. Thus, this article shows the distribution of shear-wave velocity in Ciudad del Carmen, Campeche, México which is essential for seismic characterization. Thirty MASW and ReMi measurements were made throughout the Island to obtain Vs30, the highest values were found towards the southeast, having a range of 330-400 m/s, while the lowest values are towards the northwest being 170-200 m/s, according to the soil type classifications of National Earthquake Hazards Reduction Program and International Building Code, D and E type soil predominate. The soil of the

Ciudad del Carmen was regionalized into three types, where type I being a dense soil or averagely soft rock with Vs30 greater than 360 m/s, type II when the soil has an intermediate dynamic amplifications with Vs30 between 180 to 360 m/s, and type III correspond to a soil with large dynamic amplifications and Vs30 less than 180 m/s. The structural response of buildings to earthquake shaking is affected by the interactions between three linked systems: the structure itself, its foundations and the underlying and surrounding soil or rock, so it is important mapping the subsurface, to obtain the shear-wave velocities of the

subsoil in the study area to evaluate soil-structure interaction or seismic base isolation of buildings. Field results show the critical role that shear-wave velocities have for the structural design of buildings in Ciudad del Carmen. Further research is warranted on the role of shear-wave velocities to conduct a soil-structure interaction in the design guidelines of Carmen municipality.

## Author contribution

**Leonardo Palemón-Arcos**: Resources, Field experiments, Conceptualization, Methodology, Writing-Original draft preparation. **Carmen M. Gómez-Arredondo**: Field experiments, data acquisition and analysis, Conceptualization, Formal analysis, Writing-Original draft preparation. **Daniel A. Damas-López**: Field experiments, Data acquisition and analysis,





Software, Visualization. **G. Chávez-Hernández**: Field experiments, Data acquisition and analysis, Software. **Yuriko Gutiérrez-Can**: Formal analysis, Software, Writing-Original draft preparation. **Marco A. Hernández-Hernández**: Field experiments, Data acquisition and analysis, Software. **Edén Bojórquez**: Conceptualization, Formal analysis, Software, Writing-Original draft preparation. **Barrera-Lao**: Field experiments, Data acquisition and analysis, Visualization.

**Competing interests**

The authors declare no conflicts of interest to disclose. In a disaggregated way, we wish to confirm that there are no known conflicts of interest associated with this publication and there has been no significant financial support for this work that could have influenced its outcome. We confirm that the manuscript has been read and approved by all named authors and that there are no other persons who satisfied the criteria for authorship but are not listed. We further confirm that the order of authors listed in the manuscript has been approved by all of us. We confirm that we have given due consideration to the protection of intellectual property associated with this work and that there are no impediments to publication, including the timing of publication, with respect to intellectual property. In so doing we confirm that we have followed the regulations of our institution concerning intellectual property.

**Data availability**

The data that support the findings of this study are openly available.

**Acknowledgments**

The first author conducted this study at the Autonomous University of Carmen with the support of the Conacyt SNI program. We acknowledge special the field support provided by Angelita Figueroa, Yatzareli Alonso, Elioenai Balcazar, Néstor Canché, Gerardo Herrera, Carmen Leonardo Vázquez, Héctor Juárez, Juan Osvaldo Vázquez and Juan Angel Exzacarias.

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





**Table 1. Site classification ASCE/SEI 7–16 (ASCE, 2017; BSSC, 2009; ICC, 2017)**

| Site Class | $\overline{Vs}$ (m/s) | $\overline{N}$ or $\overline{Nch}$ | $\overline{Su}$ (kPa) |
|---|---|---|---|
| **A** Hard rock | > 1,520 | NA | NA |
| **B** Rock | 760 to 1,520 | NA | NA |
| **C** Very dense soil and soft rock | 365 to 760 | > 50 blows /ft | > 96 |
| **D** Stiff soil | 182 to 365 | 15-50 blows /ft | 48-96 |
| **E** Soft clay soil | < 182 | < 15 blows /ft | < 48 |
| | Any profile with more than 3 m of soil that has the following characteristics: | | |
| | — Plasticity index PI > 20, <br> — Moisture content w ≥ 40 %, <br> — Undrained shear strength, $\overline{Su}$ < 25 kPa. | | |
| **F** Soils requiring site response analysis in accordance with Section 21.1 | See Section 20.3.1 | | |

Note: For SI: 1 ft = 0.3048 m; 1 ft /s=0.3048 m/s; 1 lb /ft$^2$ = 0.0479 kN /m$^2$.

**Table 2. Vs30 zonation**

| Soil type | Shear-wave velocity (m/s) | Description |
|---|---|---|
| **I** | **Vs30 ≥ 360** | **Dense soil or averagely soft rock** |
| **II** | **180 ≤ Vs30 < 360** | **Soil with intermediate dynamic amplifications** |
| **III** | **Vs30 < 180** | **Soil with large dynamic amplifications** |



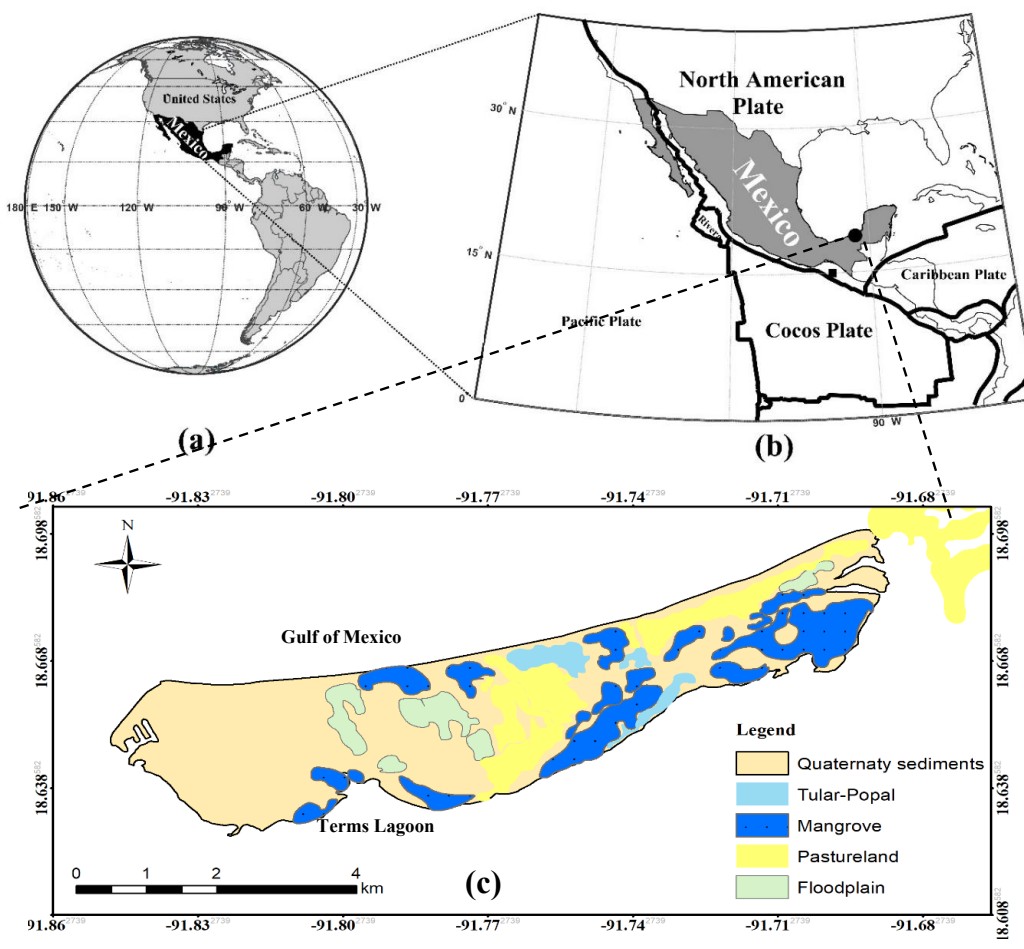

**Figure 1: Study area location, (a) Mexico in world, (b) tectonic plates of Mexico with study area (shaded circle) and Earthquake of September 8, 2017 (shaded square) and (c) surficial geology of Ciudad del Carmen, most part of study is lying by recent quaternary sediments.**


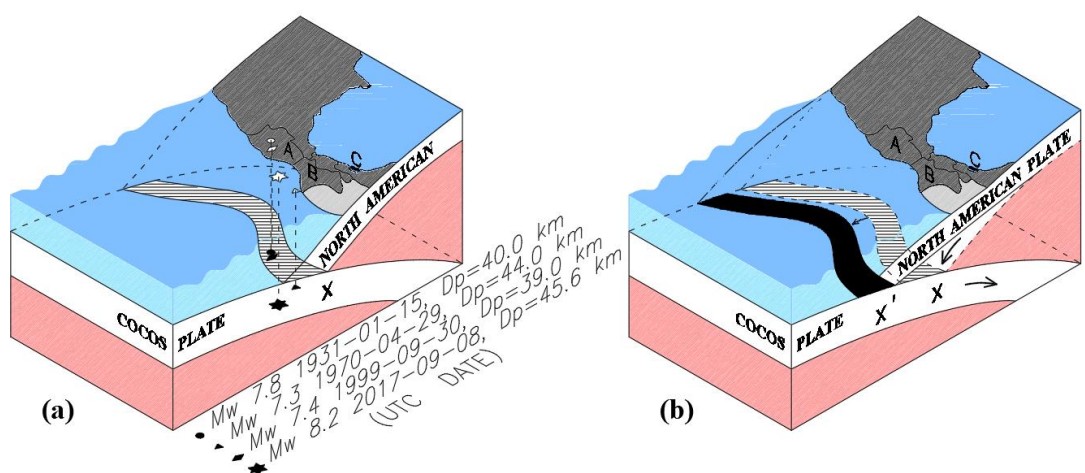

**Figure 2: Basic sketch of southwest Mexican subduction plate, (a) without movement but with representative earthquakes in point A-Oaxaca, point B-Chiapas and point C indicates Ciudad del Carmen and (b) with displacement (x to x') causing earthquakes.**
**Adapted from Palemón-Arcos *et al*., (2020).**

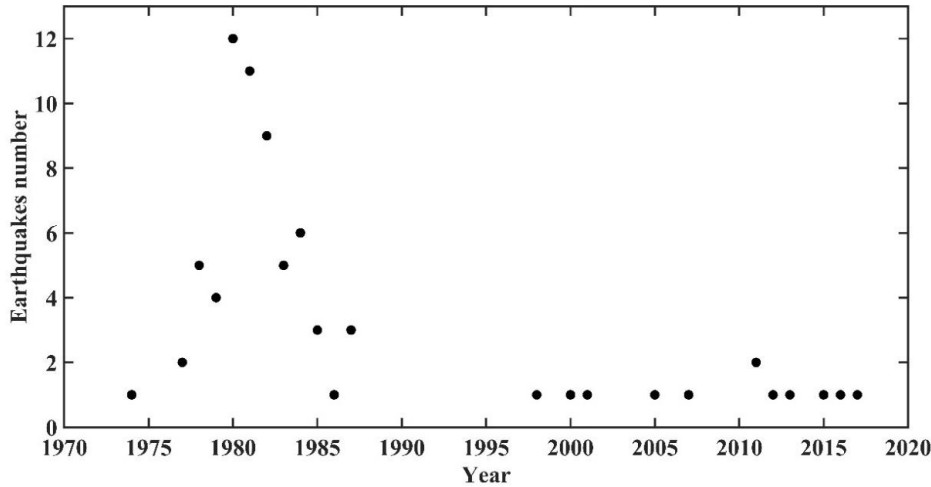

**Figure 3: Seventy-four earthquakes from 1974 to 2020 in Ciudad del Carmen of the Campeche state (SSN, 2020).**

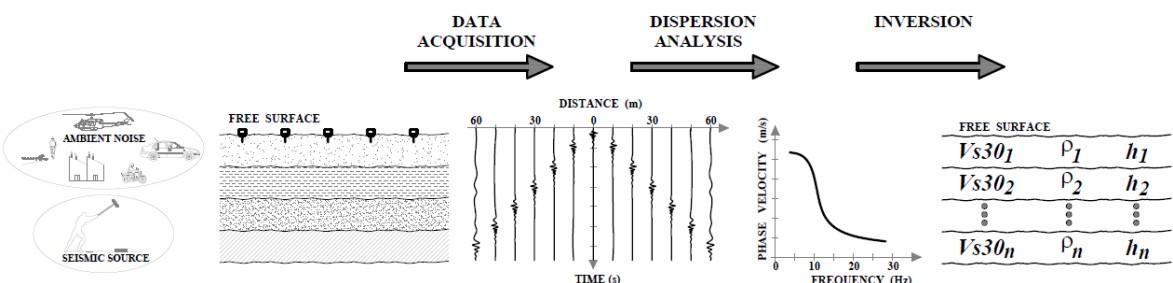

**Figure 4: Methodology framework.**


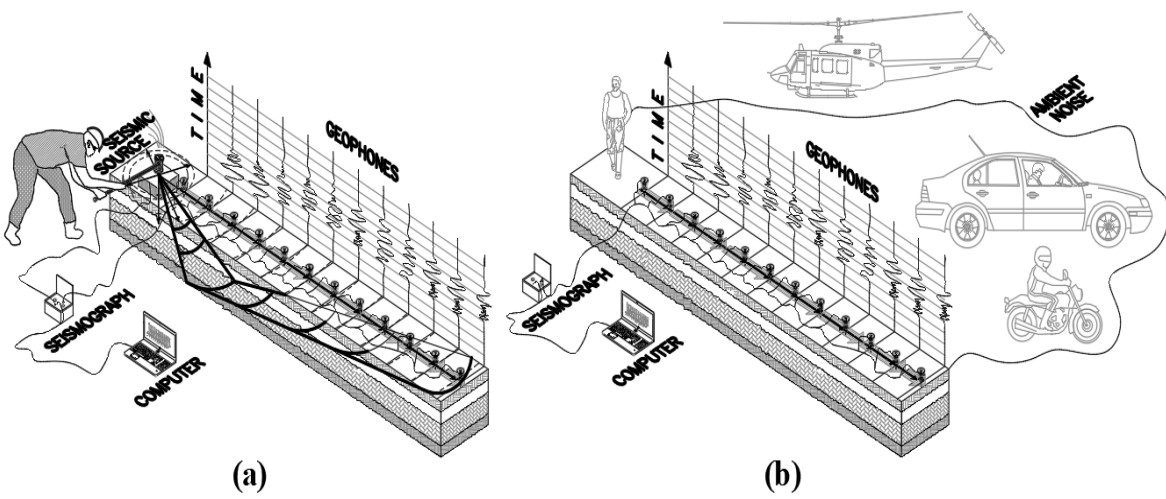

**Figure 5: Schematic diagram of shear wave travel paths for (a) active and (b) passive methods.**





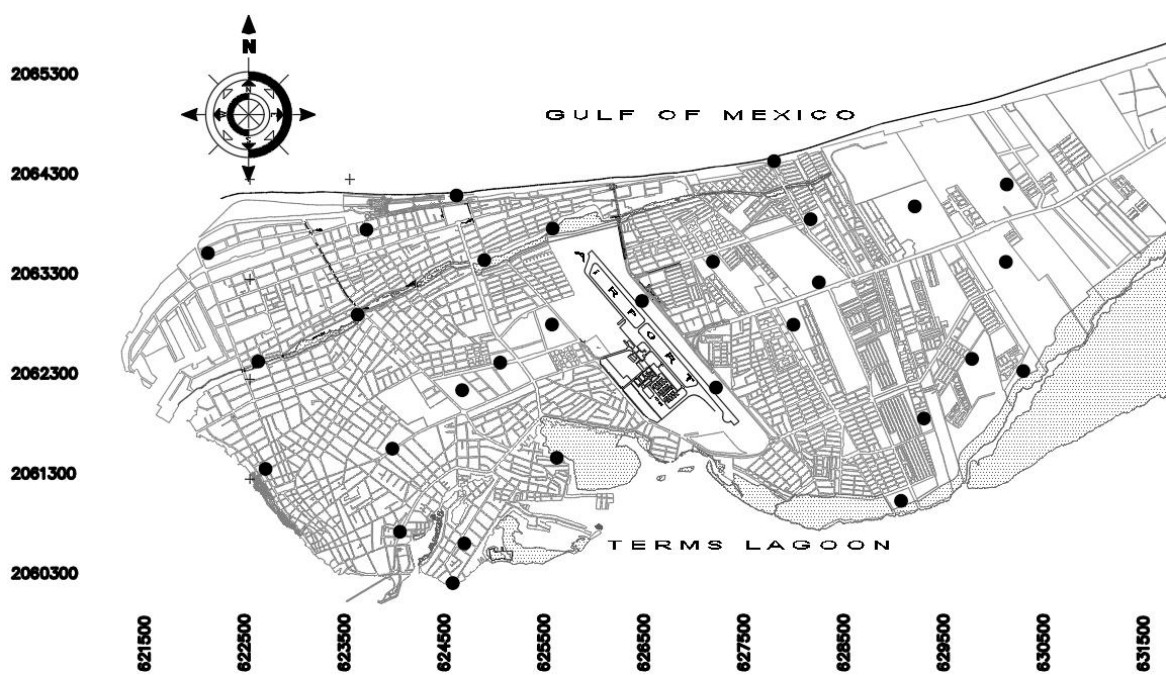


**Figure 6: Map snapshot for thirty data survey acquisition points (shaded circles) in Ciudad del Carmen.**

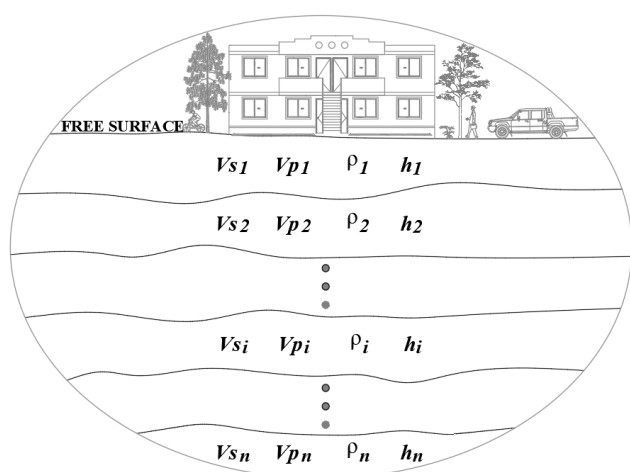

**Figure 7: Heterogeneous layers of subsoil with parameters vectors of dispersion curves where n is the number of layers. Adapted from Xia, Miller and Park (1999).**


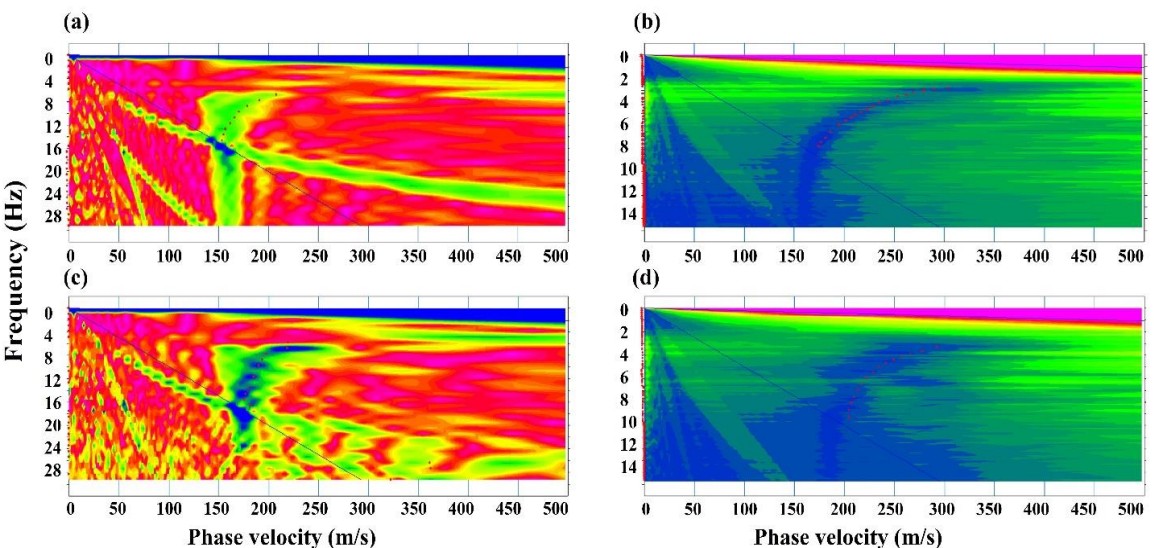

**Figure 8: Dispersion images obtained from (a) active in point 2, (b) passive in point 2, (c) active in point 4 and (d) passive in point 4 of MASW and ReMi surveys. The red points in each figure represents the maximum amplitude picked in each frequency.at velocity spectrum.**

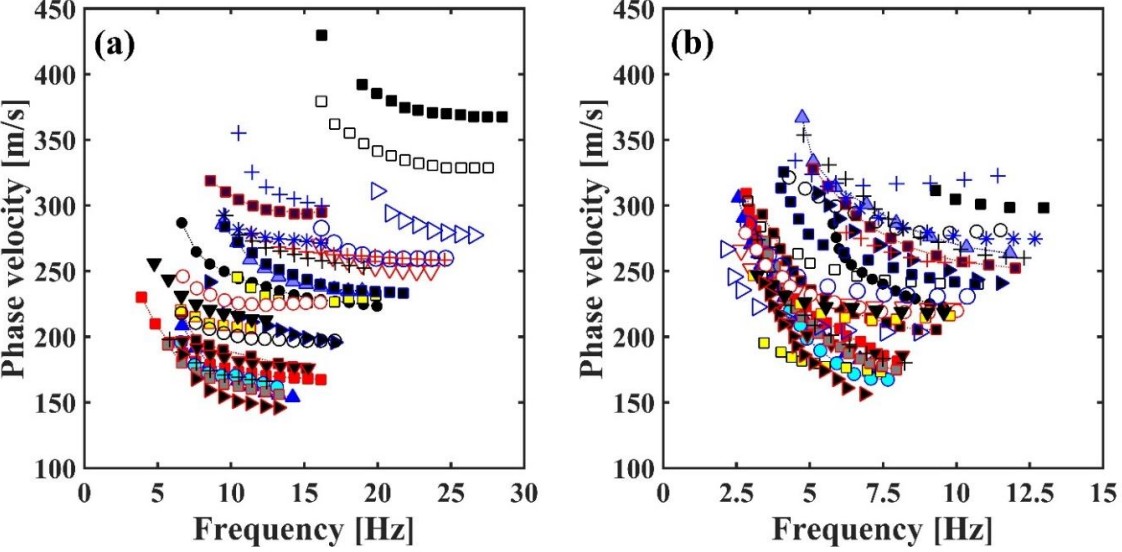

**Figure 9: Dispersion curves acquired by (a) active and (b) passive method in each survey.**




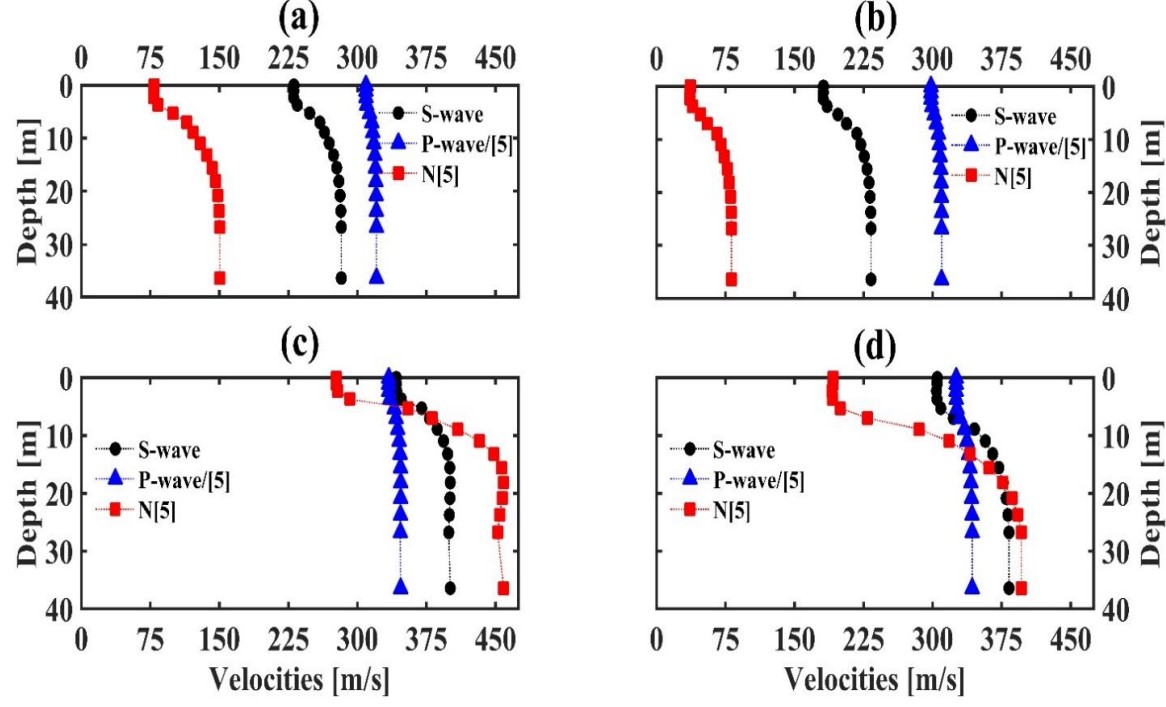

Figure 10: P and S waves velocities by active approach at (a) point 1, (b) point 4, (c) point 14 and (d) point 24.

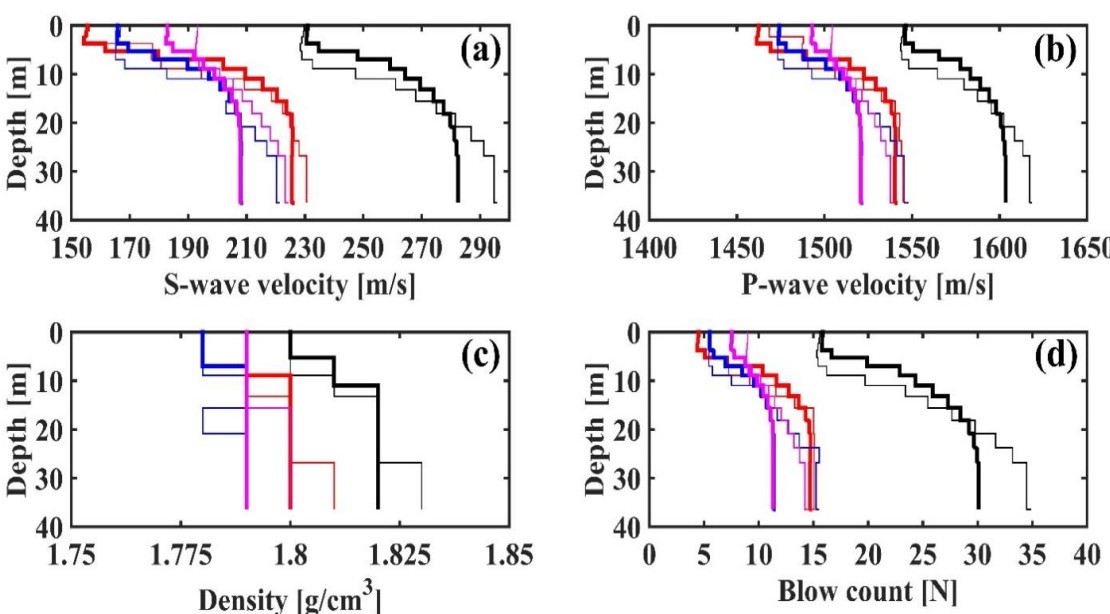

Figure 11: Comparison of results by active and passive methods (a) S-waves, (b) P-waves, (c) density and (d)blow count.

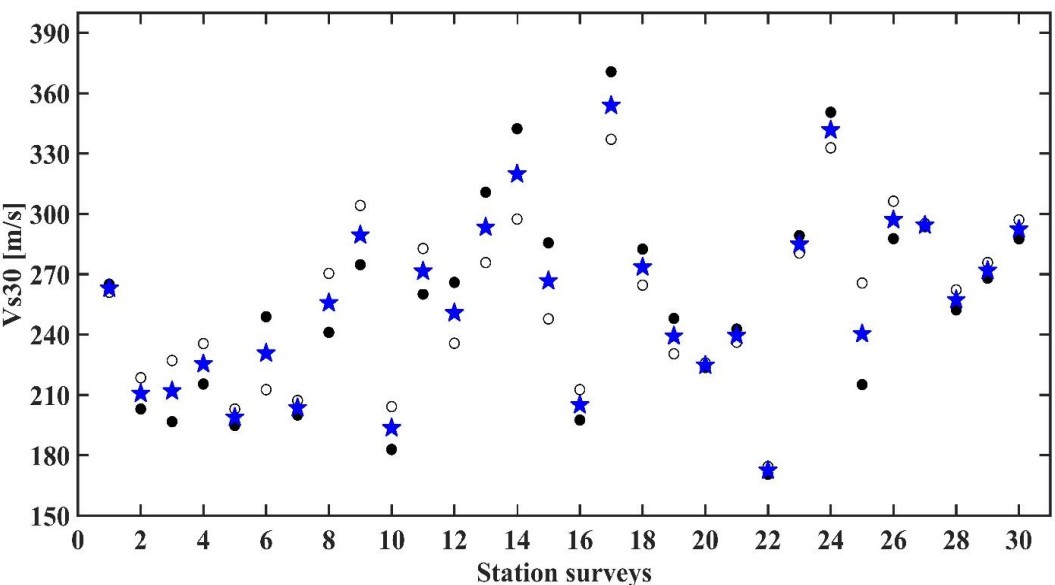

Figure 12: Shear-waves velocities comparison of results by active (shaded circles) and passive (unshaded circles) methods and the average of both (blue pentagram).



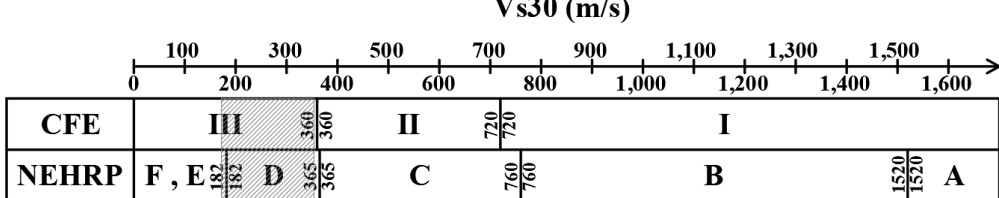

**Figure 13: Soil classification (hatched area) of Ciudad del Carmen based on CFE (CFE, 2015) and NEHRP (BSSC, 2009).**

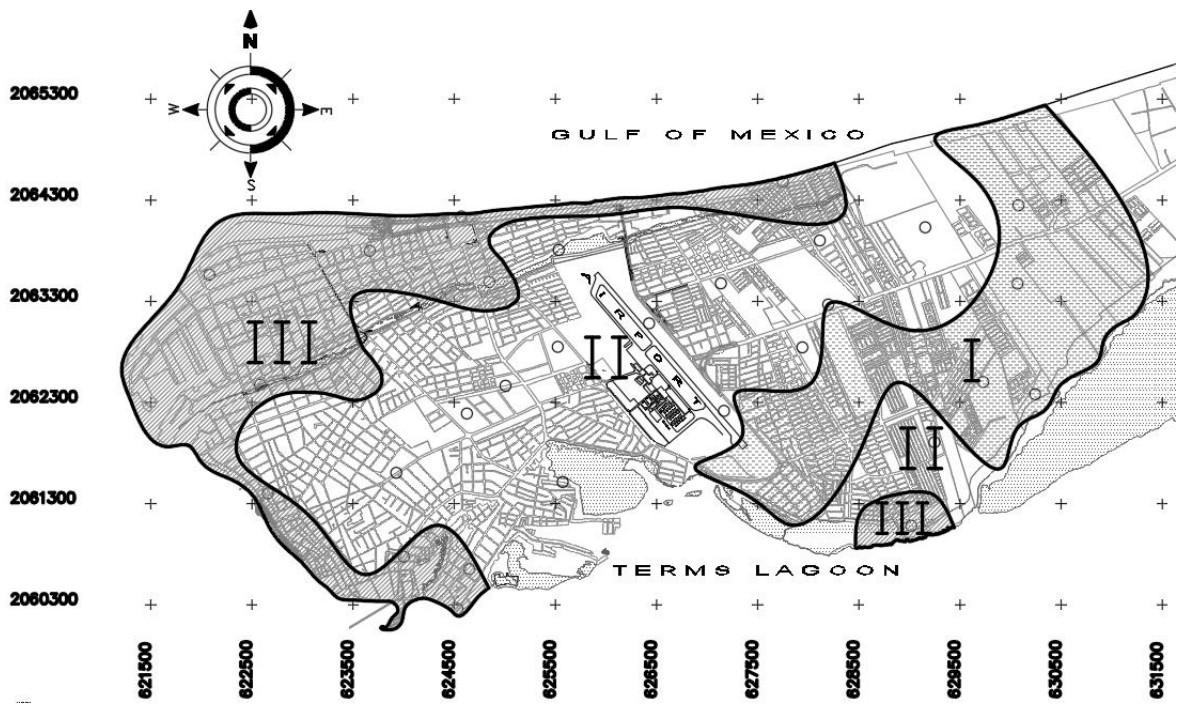


**Figure 14: Seismic zonation map from Vs30 classification for Ciudad del Carmen Campeche Mexico area.**