# Peer review of "Subsoil seismic characterization through Vs30 for future structural assessment of buildings (Ciudad del Carmen, Mexico)"

_Natural Hazards and Earth System Sciences, 2020_

## Referee Comment (RC1) · Anonymous Referee #1 · 21 Nov 2020

The paper has been written as a technical report, needs to be rewritten, and it should be clear what they want to solve. Likewise, its results should be presented in terms of a characterization of seismic site effects as a preliminary step for quantification studies of seismic hazard. Some remarks 1. An abstract summarizes, usually in one paragraph of 300 words or less, the major aspects of the entire paper in a prescribed sequence that includes: 1) the overall purpose of the study and the research problem(s) you investigated; 2) the basic design of the study; 3) major findings or trends found as a result of your analysis; and, 4) a brief summary of your interpretations and conclusions (See https://libguides.usc.edu/writingguide/abstract#:∼:text=An%20abstract%20summarizes%2C 2. The goal of the study is not clear. 3. The section Introduction must be focusing

in describe the importance of know the subsoil seismic response in areas with the absence of seismic studies and what methodologies are recommended applied. 3. Is it necessary to change the focus of the geological framework section to a seismotectonic framework only for southern Mexico. 4. Section Previous studies. The text is meaningless, in this section I recommend to authors focus on the interest area, they should describe the general context of the seismic hazard in southern Mexico, and the information available about seismic site effects. Is not necessary to describe aspects of all Mexico that not will be used later. 5. Section location. Rewrite. 6. Section and methods. Is needed to describe why you use MASV technique in this study. 7. It is necessary to make a correlation of the results with geotechnical information and geological information.

---

## Referee Comment (RC2) · Anonymous Referee #2 · 31 Aug 2021

This paper presents the seismic microzoning of Ciudad del Carmen city following the Mexican building code (CFE). Though the subject could be worth publishing, the paper should be entirely re-written.

Here are my main comments: "1-Introduction and 2-study area": It should be focused on Ciudad del Carmen context in terms of tectonics, regional seismicity, superficial geology and building code. Figures 2 and 3 should be replaced by a general map of the regional seismicity.

"3-Materials and methods": It is not necessary to explain MASW and REMI methods

since they are well known methodologies now (Figures 5 and 7 could be removed). Same remark for dispersion analysis. On the contrary, it should be interesting to detail the local context of Ciudad del Carmen in terms of population, vulnerability and seismic hazard.

"4-Numerical results": Results are difficult to comment. If the aim of the paper is to present a microzoning of the city, MASW profiles should be clearly identified in Figure 6, 9, 10 and 11. The authors should comment the results profiles by profiles and compared them to the superficial geology and the expected soil response. It is not clear for me if the final zonation is based on the NEHRP building code or the CFE one. Is there a specific building code in Ciudad del Carmen? Why the authors choose to divide the city in 3 zones (I, II and III) while the VS30 values give only a soil type III (Figure 13)?

General comment on the paper: the English language needs to be reviewed. The text is poorly argued and the discussion is not precise enough to validate the results. Some figures could be removed (2, 5, 7, 8, 11) and others should be highly improved.

---

## Author Comment (AC1) · 12 Oct 2021

Response to comments of Referee #1 "Subsoil seismic characterization through Vs30 for future structural assessment of buildings (Ciudad del Carmen, Mexico)" nhess-2020-194

Comments to Author The paper has been written as a technical report, needs to be rewritten, and it should be clear what they want to solve. Likewise, its results should be presented in terms of a characterization of seismic site effects as a preliminary step for quantification studies of seismic hazard. Some remarks

1. An abstract summarizes, usually in one paragraph of 300 words or less, the major aspects of the entire paper in a prescribed sequence that includes: 1) the overall purpose of the study and the research problem(s) you investigated; 2) the basic design of the study; 3) major findings or trends found as a result of your analysis; and, 4) a brief summary of your interpretations and conclusions (See https://libguides.usc.edu/writingguide/abstract#::text=An%20abstract%20summarizes%2C%20usually%20in,as%20a%20

RESPONSE: We thank the reviewer for providing many constructive comments that allow us to improve the manuscript presentation. Furthermore, a thorough revision on the manuscript writing has been conducted. For comment number 1, this information has been included in the revised manuscript (see Page 1, Line 14 to 29). "Not all sites in a country are characterized geotechnical or seismically, especially those classified as low seismicity by the country's regulations. However, nearby earthquakes with epicenters no greater than 500 km may arise at a site of interest, for example, due to the soil type, on September 8, 2017, the instraslab Chiapas-Mexico earthquake was felt causing damage to the facades of the buildings in Ciudad del Carmen Campeche, and major structural damage in the state neighboring. Therefore, for the purpose of mitigating subsequent damage with another earthquake equal to or greater magnitude, it is important to have the subsoil shear-waves velocities as a preliminary phase for quantification studies of seismic hazard as well as analysis and design structural seismic considering soil-structure interaction and soil liquefaction. For this reason, in this study under the seismic approach, an assessment of Ciudad del Carmen Campeche subsoil is presented. Active and passive Multichannel Analysis of Surface Waves and Refraction Microtremor technique to investigate seismically subsoil characteristics have been employed. Shear wave velocities were obtained up to a depth of 30 m with magnitudes of 172.45 m/s to 353.90 m/s. Based on the Vs30 values, the subsoil is seismically classified into D and E according to the criterion of the National Earthquake Hazards Reduction Program and International Building Code, turning out to be very vulnerable to high damage during the earthquake shaking. Furthermore, Ciudad del Carmen was regionalized into three types, where type I being

a dense soil or averagely soft rock with Vs30 greater than 360 m/s, type II when the soil has an intermediate dynamic amplification with Vs30 between 180 to 360 m/s, and type III correspond to a soil with large dynamic amplifications and Vs30 less than 180 m/s".

2. The goal of the study is not clear. RESPONSE: For comment number 2, this information has been included in the revised manuscript (see Page 2, Line 39 to 41). "Therefore, the purpose of this project is to characterize the subsoil through shear wave velocities as a preliminary step for the quantification studies of the seismic hazard as well as the analysis and structural seismic design considering the soil-structure interaction and soil liquefaction".

3. The section Introduction must be focusing in describe the importance of know the subsoil seismic response in areas with the absence of seismic studies and what methodologies are recommended applied. RESPONSE: The reviewer is right, this suggestion has been incorporated in the new version of the manuscript (see lines 40-46, page 2).

4. Is it necessary to change the focus of the geological framework section to a seismotectonic framework only for southern Mexico. RESPONSE: The reviewer is right, this suggestion has been incorporated in the new version of the manuscript (see lines 94-102, page 4). Southeast Mexico is lying in a complex tectonic zone because the movements of the North American, Caribbean and Cocos plates (Fig. 1b) converge in this region from the late Oligocene (Morán Zenteno et al., 2000), that is, the North American Plate moves to the west with respect to the Caribbean, while the Cocos Plate moves to the Northeast with respect to the previous ones. The boundary between North American and Caribbean plate is driven by the Polochic Motagua fault system, in which North American plate laterally moves with respect to the Caribbean to 1.7 cm/year, while the Cocos Plate subducts under North American with a convergence rate of 7 cm/year. This regional tectonic involve an interesting subduction process in the region of Tehuantepec Ridge, a transform fault. The geometry of Cocos Plate

changes drastically from the East to West near the Tehuantepec Isthmus, the dip angle changes from ~45° to a low angle subduction of ~25° in the same direction (Manea et al., 2014).

---

## Author Comment (AC4) · 12 Oct 2021

Response to comments of Referee #2 "Subsoil seismic characterization through Vs30 for future structural assessment of buildings (Ciudad del Carmen, Mexico)"

nhess-2020-194 Comments to Author This paper presents the seismic microzoning of Ciudad del Carmen city following the Mexican building code (CFE). Though the subject could be worth publishing, the paper should be entirely re-written The reviewer is right, this suggestion has been incorporated in the new version of the manuscript. We thank the reviewer for providing many constructive comments that allow us to improve the

manuscript presentation.

1-Introduction and 2-study area": It should be focused on Ciudad del Carmen context in terms of tectonics, regional seismicity, superficial geology and building code. Figures 2 and 3 should be replaced by a general map of the regional seismicity RESPONSE: Tectonics, regional seismicity, and superficial geology has been included in the revised manuscript (see Page 4, Lines 94 to 125). Regarding the building code, there is only the construction regulations of September 8, 1997, lacking the seismic and structural part, so here is the objective of the paper to seismically regionalize the city of Carmen.

3. Materials and methods": It is not necessary to explain MASW and REMI methods The goal of the study is not clear. since they are well known methodologies now (Figures 5 and 7 could be removed). Same remark for dispersion analysis. On the contrary, it should be interesting to detail the local context of Ciudad del Carmen in terms of population, vulnerability and seismic hazard RESPONSE: The reviewer is right, observation attended. Figures were removed in the revised manuscript (see Page 3, Line 89 to 92). "As of 2010, Ciudad del Carmen had a population of 169,466 (INEGI, 2010), and has a great demand for housing due to the oil boom and accelerated population growth so the houses are built on sandy soil and with vertical growth generating a risk from natural phenomena such as: hurricanes, flooding, tropical storms and cold fronts. Additionally, it is located just 500 km from the epicenters, which brings a seismic hazard with possible sand liquefaction."

4. Numerical results": Results are difficult to comment. If the aim of the paper is to present a microzoning of the city, MASW profiles should be clearly identified in Figure 6, 9, 10 and 11. The authors should comment the results profiles by profiles and compared them to the superficial geology and the expected soil response. It is not clear for me if the final zonation is based on the NEHRP building code or the CFE one. Is there a specific building code in Ciudad del Carmen? Why the authors choose to divide the city in 3 zones (I, II and III) while the VS30 values give only a soil type III (Figure 13)?.

RESPONSE: The reviewer is right. Suggestion has been incorporated. Other answers are listed below: Another paper to obtain geomechanical properties of soil through correlations between the MASW technique and geotechnical studies is planned for Ciudad del Carmen with the conditions of the water table. For the final zonation, only the NEHRP and CFE codes were taken as reference. Regarding the building code, there is only the construction regulations of September 8, 1997, lacking the seismic and structural theme, so here is the objective of this paper, to seismically regionalize Ciudad del Carmen considering the reference velocities of the NEHRP and CFE codes. Geotechnically and seismically, with the two reference codes it is observed that there is only one type of soil. However, it is divided into three zones because there are clearly areas of soft soil product of fillings, healthy areas without fill and a third with competent soil.

General comment on the paper: the English language needs to be reviewed. The text is poorly argued and the discussion is not precise enough to validate the results. Some figures could be removed (2, 5, 7, 8, 11) and others should be highly improved RESPONSE: The reviewer is right, observation attended.